# Patients with Thyroid Disorder, a Contraindication for Dental Implants? A Systematic Review

**DOI:** 10.3390/jcm11092399

**Published:** 2022-04-25

**Authors:** Aina Torrejon-Moya, Keila Izquierdo-Gómez, Mario Pérez-Sayáns, Enric Jané-Salas, Antonio Marí Roig, José López-López

**Affiliations:** 1Department of Odontoestomatology, Faculty of Medicine and Health Sciences, School of Dentistry, University Campus of Bellvitge, University of Barcelona, 08907 Barcelona, Spain; aina.torrejon@gmail.com (A.T.-M.); keila_izqdo@hotmail.com (K.I.-G.); enjasa19734@gmail.com (E.J.-S.); 2Oral Health and Masticatory System Group, IDIBELL (Bellvitge Biomedical Research Institute), University of Barcelona, 08907 Barcelona, Spain; 3Oral Medicine, Oral Surgery and Implantology Unit (MedOralRes), School of Medicine and Dentistry, University of Santiago de Compostela, 15782 Santiago de Compostela, Spain; perezsayans@gmail.com; 4Department of Maxillofacial Surgery, Bellvitge University Hospital, L’Hospitalet de Llobregrat, 08907 Barcelona, Spain; ebusitano@gmail.com

**Keywords:** thyroid disorder, hypothyroidism, hyperthyroidism, dental implants

## Abstract

The thyroid gland is composed of the thyroid follicles, considered to be the functional units of the thyroid gland. The synthesis of the thyroid hormones occurs in these follicles. Triiodothyronine (T3) and thyroxine (T4) are the thyroid hormones and affect metabolic processes all through the body. This systematic evaluation was performed to answer the following PICO question: “Can patients with thyroid disorders undergo dental implant rehabilitation with the same survival rate as patients without thyroid disorders?”. A systematic review of the literature was conducted following the Preferred Reporting Items for Systematic Reviews and Meta-Analyses (PRISMA) statements to gather available and current evidence of thyroid disorders and its relationship with dental implants. The electronic search, in the PubMed and Cochrane databases, yielded 22 articles. Out of the 22 articles, only 11 fulfilled the inclusion criteria. Manual research of the reference list yielded no additional papers. According to the SORT criteria and answering our PICO question, level B can be established to conclude that patients with thyroid disorders can be rehabilitated with dental implants, with similar survival rates as patients without thyroid disorders. Papers with higher scientific evidence and bigger sample size should be carried out.

## 1. Introduction

The thyroid gland is composed of the thyroid follicles, considered to be the functional units of the thyroid gland [1], and the synthesis of the thyroid hormones occurs in these follicles [1,2]. Triiodothyronine (T3) and thyroxine (T4) are the thyroid hormones and affect metabolic processes all through the body [1]; they are fundamental for normal bone turnover [1,2,3]. 

In recent years, it has been acknowledged that the thyroid plays a main role in bone development and the maintenance of bone mass, alterations in thyroid hormones lead to growth abnormalities, bone loss, and increased risk of fracture [2,4].

Thyroid hormones are essential for skeletal maturation and have a crucial physiological role in the maintenance of adult bone structure and strength [5,6]. Although thyroid dysfunction has been known to represent a risk factor for bone disease, the role of thyroid hormone in the pathogenesis of osteoporosis and risk factors of fractures has been underestimated, and the underlying mechanisms are still uncertain [2,3].

Hyperthyroidism is outlined as the suppression of Thyroid-Stimulating Hormone (TSH) with increased T3 and T4, mainly caused by Graves’ disease, toxic multinodular goiter, and toxic adenoma [3]. It has a detrimental effect on bone mass due to a high bone turnover, as documented by a shortened bone remodeling cycle, together with an increase in biochemical markers of bone resorption and bone formation [3]. producing an increase in mineral apposition and formation, as well as a decrease in bone mineral density [7]. Authors such as Delitala et al. [3] concluded by stating that increased biochemical indicators of bone turnover and a modest decrease in bone mineral density may be linked to subclinical hyperthyroidism.

Hypothyroidism is defined as increased TSH together with T3 and T4 below the lower limit of the reference range, being the main causes of acquired hypothyroidism, Hashimoto’s thyroiditis, and is post-ablative due to surgery and neck irradiation and drug-induced [2,3,8]. It impairs bone turnover by reducing both osteoclastic bone resorption and osteoblastic activity [3].

Bone is a metabolically active tissue that undergoes continual osteoblastic bone production and osteoclastic bone resorption. As a result, the ability of bone tissue to adapt to damages such as implant placement is linked to several processes and can be influenced by a variety of factors [8,9] such as smoking, oral hygiene, and prosthetic rehabilitation, affecting osseointegration and reducing the success rate of dental implants [10].

In the long-term follow-up, it is reported that in patients without general pathology, the survival rate and success rate of the dental implant have achieved excellent results [8]. In addition, among patients without any oral or systemic diseases, the success rate of oral rehabilitation using dental implants is 98.8% after 3 months, 97.9% after 6 months, 97.7% after 1 year, and 97.4% after 2 to 9 years [9]. These results indicate a successful rehabilitation in patients without systemic disease, taking into consideration all the following variables: age, sex, implant location, implant diameter, implant length, implant type, bone quality, bone graft, periodontal disease status, and insertion torque [9].

If patients with thyroid disorders have a direct effect on osteoclasts, or their action on bone resorption [3], Refs. [3,8,9,10] could this influence the osseointegration of dental implants, since no osseointegration indicates a low dental implant survival rate [8,9,10]? 

This systematic evaluation was performed to answer the following PICO question: “Can patients with thyroid disorders (P) undergo dental implant rehabilitation (I) with the same survival rate (O) as patients without thyroid disorders (C)?”.

## 2. Materials and Methods

A systematic review of the literature was conducted following the Preferred Reporting Items for Systematic Reviews and Meta-Analyses (PRISMA) statements [11] (Figure 1) to gather available and current evidence of thyroid disorders and their relationship with dental implants. The review was carried out from March 2021 to September 2021. Electronic research without restriction dates was carried out using three different electronic databases: PubMed, the Cochrane Central Register for Controlled Trials, and Scopus. Registration on the PROSPERO database was obtained (code: CRD42021276574).

The following terms were searched in PubMed, Cochrane, and Scopus: ““hypothyroida”“ OR““hypothyroidi”“ OR”“hypothyroidis”“[MeSH Terms] OR”“hypothyroidis”“ OR”“hypothyroi”“ OR”“hypothyroidism”“ OR”“hypothyroid”“ OR”“hyperthyroida”“ OR”“hyperthyroidi”“ OR”“hyperthyroidis”“[MeSH Terms] OR”“hyperthyroidis”“ OR”“hyperthyroi”“ OR”“hyperthyroid”“ OR”“hyperthyroidism”“)) AND ““dental implant”“[MeSH Terms] OR”“denta”“ AND”“implant”“) OR““dental implant”“).

Inclusion criteria were articles written in English or Spanish that were randomized-control trials, cohort studies, case–control studies, observational studies, and case series. On the other hand, exclusion criteria were animal studies, in vitro studies, descriptive reviews, and case reports. We also excluded patients with other systemic diseases that could influence the survival of the dental implant, such as diabetic patients, patients with bisphosphonates treatment, or other metabolic diseases. 

The primary outcome of this article was to establish whether patients with thyroid disorders had the same dental implant survival rate as patients without thyroid disorders. 

The following data were extracted from the included studies (when available): authors, year, study design, number of subjects, gender, age, thyroid disorder, number of implants, survival rate, and follow-up (in months). 

The selected studies were assessed following the Strength of Recommendation Taxonomy (SORT) criteria [12].

The risk of bias was assessed and a risk-of-bias plot (Figure 2) was created using robvis tool [13].

## 3. Results

The electronic search, in the PubMed and Cochrane databases, yielded 22 articles. Out of the 22 articles, only 11 fulfilled the inclusion criteria. 

Manual research of the reference list yielded no additional papers. 

As displayed in Table 1, the most recent article [14] was level 3 according to the SORT criteria [12], and the rest of the articles evaluated [15,16,17,18,19,20,21,22,23,24] were level 2. None of the articles were considered level 1. 

From the 11 studies, 8 were cohort studies [15,16,19,20,21,22,23,24], of which 6 were retrospective studies [15,16,19,20,22,23], 2 were prospective studies [21,24], and 2 were observational studies [17,18], which was a longitudinal study and 1 case series [14].

A total of 1111 patients were evaluated, although 3 articles [17,22,24] did not report how many thyroid patients were evaluated, with a total of 3420 placed dental implants. Again, in several articles [21,22,24], it was not reported how many implants were placed in patients with thyroid disorders. None of the articles reported guided bone regeneration before or while placing the dental implants. 

The age and gender of the patients were evaluated in a few articles, a mean age of 53 years old was calculated, and gender was evaluated in 3 articles [14,19,23], with a total of 67 (29.77%) male patients and 158 females (70.22%) patients. 

The type of thyroid disorder was evaluated in all the articles, except for Dalago HR et al. [18], which did not specify which thyroid disorder was being evaluated. Hypothyroidism was evaluated in 90.9% of the articles [14,15,16,17,19,20,21,22,23,24] and hyperthyroidism in 45.45% of the articles [19,20,21,22,24].

The medication that patients were prescribed was only mentioned by Al-Hindi M et al. [14], and patients were prescribed different doses according to their disorder. The reported medication was 75 mg of thyroxine during the week and 100 mg during the weekend, 100 mg daily, 60 mg daily, and 75 mg daily.

The implant survival rate was evaluated in most of the articles [14,18,19,20,21,22,23,24], with a mean dental implant survival rate of 92.56% in patients with thyroid disorders rehabilitated with dental implants. Alsaadi G et al. [20,21] evaluated the dental implant survival rate differentiating between hyperthyroidism and hyperthyroidism. The mean implant survival rate for patients with hyperthyroidism evaluated in the two articles [20,21] was 93.18% and 96.84% in hypothyroidism. 

Ursomanno BL et al. [15], instead of evaluating the dental implant survival rate, estimated a marginal bone loss of 0.53 mm/year in patients with hypothyroidism disorder. 

Additionally, the follow-up was mentioned in three articles [14,19,23]: Al-Hindi M et al. [14] described a follow up of 6 to 12 months after loading, De Souza JG et al. [19] had a follow-up of up to 105 months, and Attard NJ et al. [23] 1 to 20 years of follow-up.

## 4. Discussion

The present study aimed to investigate any possible association between thyroid disorders and the survival of dental implants, based on the published data.

In agreement with Diab N et al. [25], our systematic review also showed that middle-aged women are a high-risk group of thyroid diseases, showing a clear prevalence for the older population and women. 

Although the survival rate was only evaluated in nine papers [9,13,14,15,16,17,18,19], the mean implant survival rate was 92.56%, which is similar to the implant rehabilitation survival rate of patients without any systemic condition, which ranges from 92% to 95%, depending on the implant prosthetic rehabilitation [26]. We know there are several factors related to the prosthetic rehabilitation such as the type of implant–prosthetic connection, the morphology and material of the abutment, the design and material of the screw, tolerances between the screw and thread, the morphology of the implant fixture, and the type of prosthetic rehabilitation [27], but these were not evaluated in the reviewed articles; therefore, they were not analyzed in the systematic review. This is one of the main limitations of our study. 

Ursomanno BL et al. [15] evaluated the marginal bone loss, concluding with 0.53 mm/year in patients with hypothyroidism disorder. This result can be compared with patients without any systematic disease as Saravi E et al. [28] stated: 0.17 ± 0.07 mm to 2.1 ± 1.6 mm in fixed rehabilitations and from 0.22 ± 0.55 mm to 2.5 ± 2.7 mm in removable rehabilitations. Again, it has been stated that marginal bone loss can depend on different factors such as the thickness of the peri-implant soft tissue [29], heavy smoking, or bisphosphonates therapy [30].

However, we would like to highlight the fact that only one article [14] stated the medication that patients ingested, and the follow-up was only mentioned in three articles [14,19,23]. None of the articles reported if the patients had a bad or good control of the disease and for how long they had been diagnosed with the thyroid disorder. For this reason, the results of this systematic review should be interpreted with caution. Therefore, more studies that include medication, the diagnosis of the thyroid disorder, and a general evaluation of the patient are mandatory because should medicated and controlled patients not be treated with the same risk factors as patients without any other medical condition?

On the other hand, papers with higher scientific evidence and bigger sample sizes should be carried out. 

De Souza JG et al. [19] reported a dental implant survival rate of 71.2%, being the lowest one reported, in their study. They attributed this rate to the prosthetic rehabilitation and the history of periodontitis, not the thyroid disorder. Although this result cannot be compared to the other studies [14,16,17,18,20,21,22,23,24], none of them presented an association between the results and the prosthodontic rehabilitation or the history of periodontitis. For instance, other factors that have been related to a low survival rate are the type of implant surface (smooth versus rough) or the placement of a dental implant in a retreated area [31].

In addition, we think it is relevant to outline that none of the evaluated studies reported performing guided bone regeneration (GBR) in order to place the dental implants. Since some studies reported a higher marginal bone loss on implants with GBR compared to those without GBR [32], we think it should be and evaluated parameter, in the interest of having a more homogeneous sample.

Due to the heterogenicity of the studies, we were not able to perform a meta-analysis regarding the survival rate. 

## 5. Conclusions

Answering our PICO question, level B can be established to conclude that patients with thyroid disorders can be rehabilitated with dental implants, with similar implant survival rates as patients without thyroid disorders, even though more studies with larger sample sizes and higher levels of evidence, such as randomized controlled trials, are needed.

## Figures and Tables

**Figure 1 jcm-11-02399-f001:**
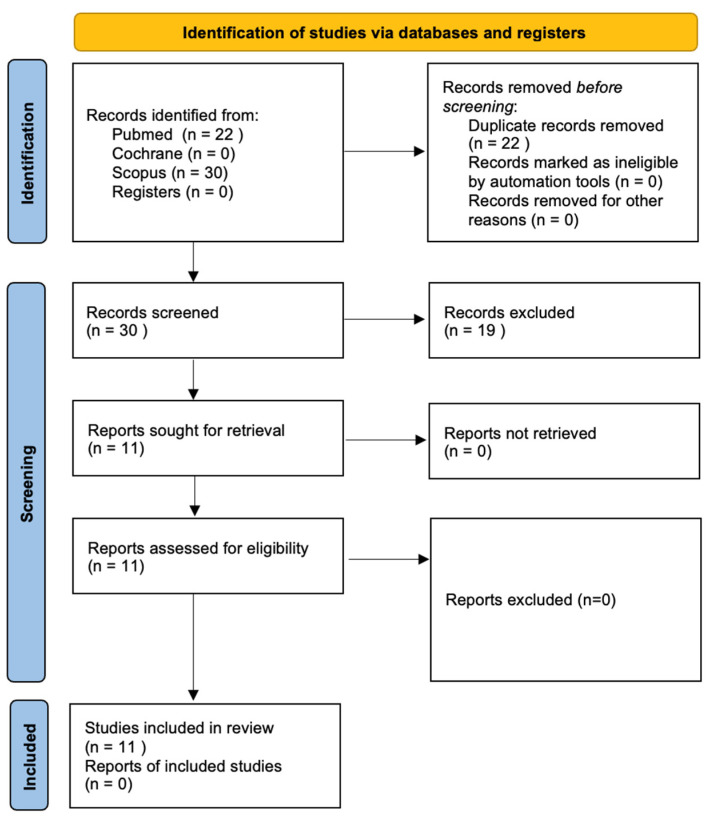
PRISMA Flow Diagram.

**Figure 2 jcm-11-02399-f002:**
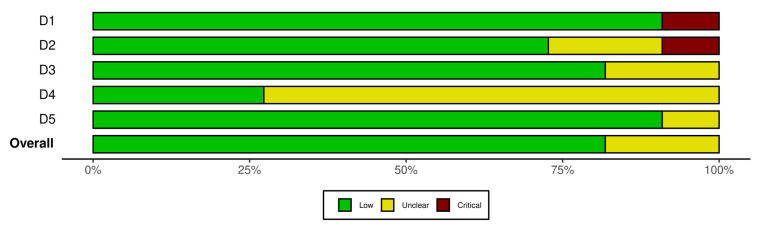
Risk-of-bias plot.

**Table 1 jcm-11-02399-t001:** Summary of the studies evaluated.

Article	Study Design (SORT)	Number Subjects	Gender (Mean Age)	Thyroid Disorder	Medication	Number of Implants	Survival Rate	Bone Loss (mm/Year)	Follow-Up
Al-Hindi M et al. (2021) [14]	Review Case series (3)	5	5 F (38.4)	Hypo	75 mg of thyroxine 75 mg during the week and 100 mg during the weekend 100 mg daily 60 mg daily 75 mg daily	16	100%		6–12 months after loading
Ursomanno BL et al. (2020) [15]	Retrospective (2)	635		Hypo		1480		0.53	
Parihar AS et al. (2020) [16]	Retrospective (2)	12		Hypo		14			
Pedro REet al (2017) [17]	Longitudinal (Observational) (2)		(71.05)	Hypo		57			
Dalago HR (2017) [18]	Cross sectional (Observational) (2)	183		Thyroid disorder (NS)		916	86.32%		
De Souza JG et al. (2013) [19]	Retrospective (2)	193	67 M and 126 F (50.3)	Hyper/hypo		722	71.2%		105 months
Alsaadi G et al. (2008) [20]	Retrospective (2)	25		Hypo		111	93.69%		
		6		Hyper		22	86.36%		
Alsaadi G et al. (2008) [21]	Prospective (2)	21		Hypo			100%		
		4		Hyper			100%		
Alsaadi G et al. (2007) [22]	Retrospective (2)			Hypo/hyper					
Attard NJ et al. (2002) [23]	Retrospective (2)	27	27F	Hypo		82	95.49%		1–20 years
Van Steenberghe D et al. (2002) [24]	Prospective (2)			Hypo/hyper			100%		
		1111	67M 158 F (53.25)	Hypothyroidism 90.9% Hyperthyroidism 45.45%		3420	92.56%		

F: Female; M: Male; Hypo: Hypothyroidism; Hyper: Hyperthyroidism; NS: Not specified.

## Data Availability

Not applicable.

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
