# Peer review of "Patients with Thyroid Disorder, a Contraindication for Dental Implants? A Systematic Review"

_jcm, 2022, doi:10.3390/jcm11092399_

Round 1

Reviewer 1 Report

Dear authors,

This is an interesting review about patients with thyroid disorder and dental implants. The results, in similar studies, suggest that medically controlled hypothyroid/hyperthyroid patients are not at higher risk of implant failure than matched controls.

The search terms were few and limited (line 97-98 “The following terms were searched in PubMed, Cochrane and Scopus; “(Hypothyroidism OR Hyperthyroidism) AND Dental Implants”), which led to a very limited final selection of articles (11) for a systematic review.

  • Important issue to improve: increasing the consistency of the discussion... it is very poor. Should be further developed and with a greater degree of comparison of all important aspects contained in the selected articles.
  • more articles should be included in the final manuscript selection.

Author Response

Dear Professor,

Thank you very much for considering our paper to be published in your prestigious journal.

Below, we answer all the questions that the reviewer indicated.

Reviewer 1:

“The search terms were few and limited (line 97-98 “The following terms were searched in PubMed, Cochrane and Scopus; “(Hypothyroidism OR Hyperthyroidism) AND Dental Implants”), which led to a very limited final selection of articles (11) for a systematic review.”

# Thank you very much for your assessment, the research terms have been extended, although we weren’t able to find more articles that fitted the inclusion and exclusion criteria.

The following terms were searched in PubMed, Cochrane and Scopus; “"hypothyroida”" OR“"hypothyroidi”" OR“"hypothyroidis”"[MeSH Terms] OR“"hypothyroidis”" OR“"hypothyroi”" OR“"hypothyroidism”" OR“"hypothyroid”" OR “"hyperthyroida”" OR“"hyperthyroidi”" OR“"hyperthyroidis”"[MeSH Terms] OR“"hyperthyroidis”" OR“"hyperthyroi”" OR“"hyperthyroid”" OR“"hyperthyroidism”")) AND “"dental implant”"[MeSH Terms] OR “"denta”" AND“"implant”") OR“"dental implant”"). (Line 101)

“Important issue to improve: increasing the consistency of the discussion... it is very poor. Should be further developed and with a greater degree of comparison of all important aspects contained in the selected articles.”

# This information has been expanded and rewriten.(Line 407)

Reviewer 2 Report

Excellent work, please add risk of bias assessment chart / graph. 

Author Response

Dear Professor,

Thank you very much for considering our paper to be published in your prestigious journal.

Below, we answer all the questions that the reviewer indicated.

“Excellent work, please add risk of bias assessment chart / graph.”

# Thank you very much for your assessment, the graph has been added (Page 4).

Reviewer 3 Report

Introduce

1. The introduction is incomplete, and the impact and mechanism of thyroid disease should be explained more.

2. The evaluation of dental implant surgery should be explained more. For example, how much bone density is needed.

3. Too many paragraphs and need to be rewritten.

4. The correlation between thyroid disease and dental implant surgery should be supplemented. BMD alone is not enough to support correlation.

5. It affects the healing and density of human bones, not just thyroid disease. Such as diabetes, periodontitis, cardiovascular disease, etc. How does the author explain it?

6. [This systematic evaluation was performed to answer the following PICO question: "Can patients with thyroid disorders undergo dental implant rehabilitation with the same survival rate as patients without thyroid disorders?".]
I think more literature is needed to support the correlation of the two.

Materials and Methods

1. The literature collected in this study is too small.

2. The flow chart of the literature collection is missing.

Discussion

1. The results and discussions of the manuscript appear to be too few. Also, I didn't "really" see what was discussed. Just stating the results.

2. The discussion part is very vague, and you don't understand what the author wants to express at all? Is thyroid disorder related to dental implant surgery?

Conclusions

The author did not express the correct conclusion. I think many parts of the manuscript are missing. Not rigorous enough and should be rewritten.

Author Response

Dear Professor,

Thank you very much for considering our paper to be published in your prestigious journal.

Below, we answer all the questions that the reviewer indicated.

“The introduction is incomplete, and the impact and mechanism of thyroid disease should be explained more.”

# This information has been expanded.(Line 59 - 66)

“The correlation between thyroid disease and dental implant surgery should be supplemented. BMD alone is not enough to support correlation.”

# This information has been expanded.(Line 86 - 88)

“It affects the healing and density of human bones, not just thyroid disease. Such as diabetes, periodontitis, cardiovascular disease, etc. How does the author explain it?”

# This information has been expanded.(Line 223)  

Materials and Methods

“The literature collected in this study is too small.”

# Thank you very much for your assessment, the research terms have been extended, although we weren’t able to find more articles that fitted the inclusion and exclusion criteria.

  1. The flow chart of the literature collection is missing.

# The Flow chart has now been added (Page 3)

Discussion

  1. The results and discussions of the manuscript appear to be too few. Also, I didn't "really" see what was discussed. Just stating the results.

“The discussion part is very vague, and you don't understand what the author wants to express at all? Is thyroid disorder related to dental implant surgery?”

 # This information has been expanded.(Line 414-423)  
Conclusions

The author did not express the correct conclusion. I think many parts of the manuscript are missing. Not rigorous enough and should be rewritten.

 # This information has been rewritten.(Line 495)

Reviewer 4 Report

I have read the Guidelines for authors for Journal of Clinical Medicine and this manuscript is not appropriate for the journal.  A dental journal would be more appropriate. The idea of a systematic review of the literature to determine if patients with thyroid disease (hypo of hyper) have dental implant success rates similar to patients without thyroid disease is of little to moderate interest. Obviously, the authors did a lot of work.  However, good clinical studies of patients with thyroid disease and implants seems to be lacking and there not much can be concluded for this systematic review.

the grammar needs to be improved.  In several instances (lines 103 - 104) the authors confuse patient survival with implant survival.

There are too many short paragraphs with one or two sentences.  Commas ( ,) are used where periods should be placed.  For example, line 127 what does 53'2 mean.  Is it 53 years and two months or 53.16 years (2/12 = .166). Lines 128 - 129 - 29,77% should be 29.77 %, 70,22% should be 70.22%.

Author Response

Dear Professor,

Thank you very much for considering our paper to be published in your prestigious journal.

Below, we answer all the questions that the reviewer indicated.

“The grammar needs to be improved.”

# Thank you very much for your assessment, the text has been read by a native English-language colleague and some errors have been corrected. 

“In several instances (lines 103 - 104) the authors confuse patient survival with implant survival.”

# This information has been modified in several paragraphs.(Line 228, 269, 271, 272, 274)

There are too many short paragraphs with one or two sentences. 

# The text has been read by a native English-language colleague and some errors have been corrected. 

Commas ( ,) are used where periods should be placed.  For example, line 127 what does 53'2 mean.  Is it 53 years and two months or 53.16 years (2/12 = .166). Lines 128 - 129 - 29,77% should be 29.77 %, 70,22% should be 70.22%.

# This information has been rewritten in several paragraphs.

Reviewer 5 Report

Dear Authors,

I congratulate with you for the article.

Here my review:

The article is well written and the systematic review is correctly conducted.

- Line 86: Please add the letter P-I-C-O with parentheses after the subject they are linked. example: can patient with thyroid disorders (P) undergo....

- Please add in the discussion the reason why you didn't performed a meta analysis of the survival rates.

- the inclusion/exclusion criteria are large. Please describe them in more details. For example, did you consider also patients with others medical conditions? or did you include only patients with thyroid disorders and patient with thyroyd disorders and other systemic conditions were excluded?did you consider the type of implant insertion and load (one stage/two stage, immediate loading, etc..)? if parameters like these one were not consider, please describe them better in the inclusion criteria (it means you included patients with thyroid disorders even if they had other medical conditions) and then discuss this point in the discussion.

- Following the previous point, line 74 ''Factors such as smoking, oral hygiene and the prosthetic rehabilitation, affect osseointegration and reduce the success rate of dental implants [6]''. These phrase needs to have more support from the references. 

I suggest you a recent article which analyses some prosthetic risk factors that can be cited in order to help the reference improvement. 

Pozzan, M.C.; Grande, F.; Mochi Zamperoli, E.; Tesini, F.; Carossa, M.; Catapano, S. Assessment of Preload Loss after Cyclic Loading in the OT Bridge System in an “All-on-Four” Rehabilitation Model in the Absence of One and Two Prosthesis Screws. Materials 2022, 15, 1582. https://doi.org/10.3390/ma15041582

Author Response

Dear Professor,

Thank you very much for considering our paper to be published in your prestigious journal.

Below, we answer all the questions that the reviewer indicated.

- Line 86: Please add the letter P-I-C-O with parentheses after the subject they are linked. example: can patient with thyroid disorders (P) undergo....

# The letters have been added (Line 89)

- Please add in the discussion the reason why you didn't performed a meta analysis of the survival rates.

# This concept has been added in the discussion (Line 218)

- the inclusion/exclusion criteria are large. Please describe them in more details. For example, did you consider also patients with others medical conditions? or did you include only patients with thyroid disorders and patient with thyroyd disorders and other systemic conditions were excluded?did you consider the type of implant insertion and load (one stage/two stage, immediate loading, etc..)? if parameters like these one were not consider, please describe them better in the inclusion criteria (it means you included patients with thyroid disorders even if they had other medical conditions) and then discuss this point in the discussion.

# This information has been added in the (Line 491)

- Following the previous point, line 74 ''Factors such as smoking, oral hygiene and the prosthetic rehabilitation, affect osseointegration and reduce the success rate of dental implants [6]''. These phrase needs to have more support from the references. 

I suggest you a recent article which analyses some prosthetic risk factors that can be cited in order to help the reference improvement. 

# The suggested article has been added and mentioned. (Line 415)

Round 2

Reviewer 3 Report

1. Figure 1. PRISMA Flow Diagram 

Very true, so as to understand the design of the experiment.

2. Abstract "More studies with larger sample size and higher levels of evidence, such as randomized controlled trials, are needed."

This sentence is too similar to the conclusion, please modify it.

3. The problem with the Introduction paragraph is still unresolved and the paragraph should be shortened.

4. References are too few, please add relevant literature from 2018-to 2022.

5. Discussion Please add more discussion about thyroid disorders, and dental implants.

Author Response

  1. Figure 1. PRISMA Flow Diagram 

Very true, so as to understand the design of the experiment.

 Thank you very much for your assessment

  1. Abstract "More studies with larger sample size and higher levels of evidence, such as randomized controlled trials, are needed."

This sentence is too similar to the conclusion, please modify it.

 This information has been modified.

  1. The problem with the Introduction paragraph is still unresolved and the paragraph should be shortened.

The introduction has been modified in order to be more brief and simplified

  1. References are too few, please add relevant literature from 2018-to 2022.

 Relevant literature has been added. See references 4 to 6, and 30-31.

  1. Discussion Please add more discussion about thyroid disorders, and dental implants.

The discussion has been extended in order to provide more information.

Reviewer 4 Report

There remains at least three instances where the English grammar is such that the study is about whether patients with thyroid disease can survive implant placement. It should be what is the survival of implants in patients with thyroid disease. 

The study design and methodology is fine. The number and quality of articles are poor.  For example, lines 143 - 147 state that of the studies used for evaluation some did not report the number of patients with thyroid disorders.  Line 160 "implant survival rate was evaluated in most studies. "  Why were these studies included if the survival rate was not evaluated.  The length and follow-up was insufficient in some studies or not mentioned at all. These studies should not included.  Studies with less than  5 years should have been excluded.

Line 194 has a comma 0,53 mm/yr.  It should be 0.53 mm/yr.

None of the papers listed whether the thyroid disease was under control. 

If anything this paper demonstrates the need for better controlled studies to evaluate the influence of hyper or hypothyroidism.

Author Response

There remains at least three instances where the English grammar is such that the study is about whether patients with thyroid disease can survive implant placement. It should be what is the survival of implants in patients with thyroid disease. 

  Thank you very much for your assessment.  This information has been modified.

The study design and methodology is fine. The number and quality of articles are poor.  For example, lines 143 - 147 state that of the studies used for evaluation some did not report the number of patients with thyroid disorders.  Line 160 "implant survival rate was evaluated in most studies. "  Why were these studies included if the survival rate was not evaluated.  The length and follow-up was insufficient in some studies or not mentioned at all. These studies should not included.  Studies with less than  5 years should have been excluded.

The articles were included but not used for the evaluation of the survival rate of dental implants. The fact that the follow-up was short is being discussed in the discussion (Line 307). We accepted all the articles, even though they were written before 2017 because of the lack of number of articles, trying to achieve the maximum data from the articles. That’s why we accepted the articles that although did not report the survival rate reported comparable values.

Line 194 has a comma 0,53 mm/yr.  It should be 0.53 mm/yr.

This information has been modified. (Line 306 and 453)

None of the papers listed whether the thyroid disease was under control. 

This is being discussed in the discussion. (Line 455)

If anything this paper demonstrates the need for better controlled studies to evaluate the influence of hyper or hypothyroidism.

Exactly. Thank you very much for your assessment.  

Reviewer 5 Report

Dear Authors,

Thank you for answering my points. 

Author Response

 Thank you very much for your assessment